# Targeting Environmental and Technical Parameters through Eco-Efficiency Criteria for Iberian Pig Farms in the *dehesa* Ecosystem †

Javier García-Gudiño [1,2], Elena Angón [3,*], Isabel Blanco-Penedo [4,5], Florence Garcia-Launay [6] and José Perea [3]

1 Animal Production, CICYTEX, 06187 Guadajira, Spain
2 Animal Welfare Program, IRTA, 17121 Monells, Spain
3 Animal Production, UCO, 14071 Córdoba, Spain
4 Department of Animal Sciencie, UdL, 25198 Lleida, Spain
5 Department of Clinical Sciences, SLU, SE-750 07 Uppsala, Sweden
6 PEGASE, INRAE, Institut Agro, 35590 Saint-Gilles, France
* Correspondence: eangon@uco.es
† This paper is a part of the Ph.D. Thesis of Javier García-Gudiño, presented at University of Córdoba.

**Abstract:** Eco-efficiency could be defined as the simultaneous ability to achieve acceptable economic results with the least possible environmental degradation. Its analysis in crop and livestock production systems has become a hot topic among politicians and scientists. Pig pasture production systems are in high commercial demand because they are associated with high quality and environmentally friendly products. This work aimed to assess the eco-efficiency of pig farms and subsequently explore the determinants of inefficiency in the *dehesa* ecosystem in the southwest of the Iberian Peninsula. Farmers from 35 randomly selected farms were interviewed to obtain farm-level data. The eco-efficiency level was calculated through a joined data envelopment analysis (DEA) and life cycle assessment (LCA) approach. Subsequently, a truncated Tobit model was applied to determine factors associated with inefficiency. The results of the research revealed that Iberian pig farms are highly eco-efficient. The estimated average eco-efficiency score is 0.919 and ranges from 0.479 to 1, suggesting that the average farm could increase its value by about 8.1%. This means that the aggregate environmental pressures could be reduced by approximately this proportion (8%) while maintaining the same input level. The determinants related to social and demographic characteristics that positively affected eco-efficiency were the number of children, while years of farm activity and educational level had a negative effect. On the other hand, farm's characteristics and the type of management, the percentage of own surface area, the percentage of livestock use, and the high proportion of pigs fattened in *montanera*, positively affected the eco-efficiency level.

**Keywords:** eco-efficiency; sustainability; Iberian pig farms; environmental impact; DEA–LCA approach

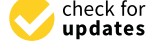



## 1. Introduction

Currently, reducing the environmental impact while maintaining a high production level has become an issue of particular interest worldwide. For this reason, numerous initiatives have been jointly launched among EU member states, such as the Green Deal, [1] the Farm to Fork Strategy [2], and the 2030 Agenda for Sustainable Development, which includes 17 sustainable development goals (SDGs) [3,4]. One of these goals (SDG 12) is "to ensure sustainable consumption and production patterns", where one of the targets is to achieve sustainable management and efficient use of sustainable resources by 2030. In this context, achieving more sustainable crop and livestock production involves bringing together different approaches within the sustainable production system and its economic,

environmental, and social pillars [5]. Concepts such as eco-efficiency, which can be defined as the simultaneous ability to achieve acceptable economic outcomes with the least possible degradation of the environment, have become a highly relevant issue in the scientific and political world [6]. Livestock activities are essential to society by supplying food, supporting rural populations and enhancing biodiversity [7]. Therefore, the search for techniques to improve the sustainability of livestock systems should be considered an essential pivotal process in all public policies at local, national, and global levels in an attempt to address the different aspects of sustainability.

A key indicator of the optimisation of resources in agricultural systems is the assessment of technical efficiency, which measures the capacity of production units to generate the maximum output level from the optimal use of resources or inputs.

On the other hand, the growing concern for cleaner products, production, and services has led organisations and companies to pursue more sustainable methods. Consequently, several methodologies have been developed to assess the environmental impact of products, such as life cycle assessment (LCA), which stands out in livestock production systems as a method to determine the environmental impact associated with production [8]. Data envelopment analysis (DEA) was also developed by Charnes et al. [9] and is widely used to estimate relative efficiency and to apply to benchmark or best practice adoption techniques [10]. It can be combined with LCA methodology in the eco-efficiency methodological framework, which is receiving significant interest as a sustainability indicator because it jointly assesses the environmental pressure of the system and the technical–economic performance of the production activity [11].

Whether at the local or national level, eco-efficiency measurement has often been used in studies of sustainability and competitiveness improvement, both at the company and sector levels. Studies stand out, especially concerning the industrial sector [12–15], the agricultural sector [11,16–20], and the livestock sector including mixed farms [4,6,7,21–24]. Environmental impact assessment studies have been carried out in Iberian pig systems [8,25–27], but there are no studies where economic results are maximised with the least possible environmental impact.

There is an area in the southwest Iberian Peninsula known as the *dehesa*, an agro-silvopastoral system based mainly on livestock farming, agriculture and forestry in areas of Mediterranean pastures. The interactions of these activities foster a high environmental value in which the combination of management decisions promotes important environmental values such as sustainable land use, a balanced landscape, and high levels of diversity at different levels of integration [26,28–30]. *Dehesa* is one of the largest managed agroecosystems in Europe representing over a million hectares. This agroecosystem is characterized by the extensive grazing of different livestock species, with the Iberian pig being the native breed most closely linked to this area [26,29–31]. The Iberian pig is reared extensively in the *dehesa* and uses natural resources (Figure 1) in the traditional fattening process based on acorns and pastures, known as the *montanera*. *Dehesa* represents the highest concentration of production and supply of the Iberian pig sector in the European Union. In recent decades, the demand for pigs in extensive production systems has grown due to the association of these production systems with high-quality and environmentally friendly meat products [32,33].

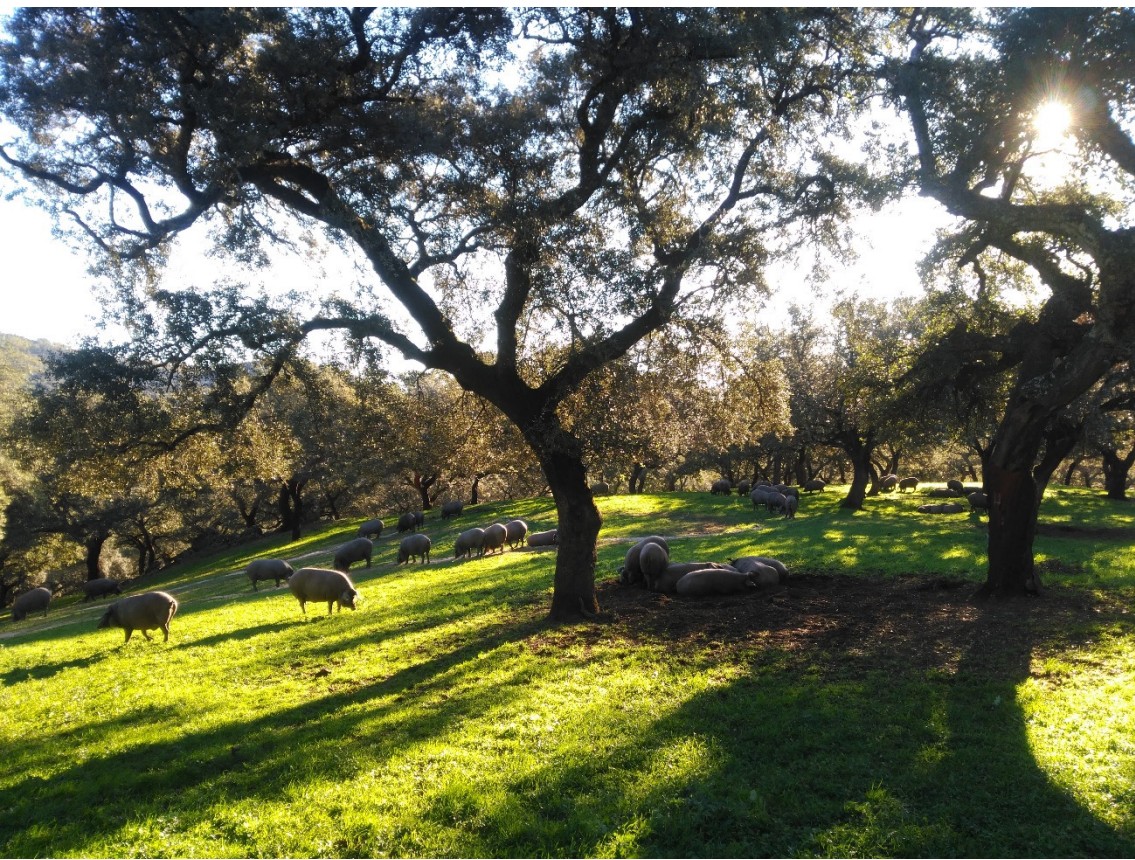

**Figure 1.** Herd of Iberian pigs reared on the *dehesa* (Source-image by Javier García-Gudiño).

Nowadays, it is a great challenge for society and for achieving sustainability to find a balance between economic performance and the use of resources. For this reason, the *dehesa* is currently suffering an alarming environmental situation due to the great stress exerted on its natural resources [26]. Therefore, it is not only essential to evaluate indicators of eco-efficiency and their production and environmental pressure reduction targets, but it is also crucial to analyse the factors that could influence the reduction of these pressures. In particular, the study at the farm level is of interest, as a large number of small-scale farms with a certain heterogeneity characterises the traditional pig sector.

This study follows an LCA–DEA approach to measure the eco-efficiency of extensive pig production in the Spanish *dehesa* and pursues two objectives, (i) using the LCA–DEA approach to calculate the level of eco-efficiency and (ii) to analyse the determinants of inefficiency using Tobit regression analysis. Pig farms' social, demographic, and structural characteristics are analysed as potential drivers of inefficiency. Understanding the key determinants that lead to inefficient production units will be beneficial for improving productivity and competitiveness and promoting a more sustainable livestock production.

## 2. Material and Methods

### 2.1. Study Area and Data Collection

The study was carried out in the traditional area of Iberian pig production, which takes place in the agroecosystem called *dehesa*. Data were collected through face-to-face questionnaires (File S1) from 35 Iberian traditional farms described in García-Gudiño et al. [8] The study data were collected during the 2016–2018 production period and include farm area, structural and productive data, economic and management aspects, information about other activities (agriculture and livestock), personal issues and labour aspects. During this period, the variables studied were not subject to inter-annual changes, and the prices of inputs and outputs remained stable.

### 2.2. LCA–DEA Approach and Tobit Model

In this study, we use a four-step approach based on an adapted version of the four-step method used by Rebolledo-Leiva et al. [34] and Angulo-Meza et al. [35]. The four-step approach optimises both pig production and environmental impact. A step-by-step schematic of the procedure is shown in Figure 2. From the data obtained (Step 1), the environmental impact of pig production on CC is estimated for each production unit or DMU (decision making unit) (Step 2) and subsequently includes it as an undesirable output in the DEA model in order to determine the environmental impact reduction (Step 3). Finally, a Tobit model is applied to analyse the eco-efficiency determinants (Step 4). It will be explained in more detail in the next sub-sections.

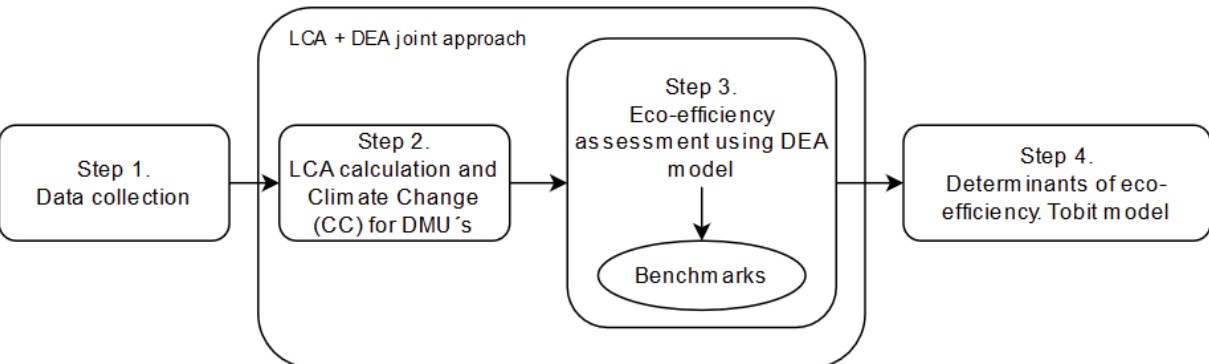

**Figure 2.** Four-step LCA–DEA and Tobit model (adapted by Rebolledo-Leiva et al. [34] and Angulo-Meza et al. [35]). Life Cycle Assessment (LCA), Data Envelopment Analysis (DEA).

#### 2.2.1. Step 1: Data Collection

As described above, data were collected on Iberian pig production systems in the *dehesa* ecosystem through personal interviews with the owners of 35 traditional Iberian farms. Due to the peculiarities of the production system, more than one visit per farm had to be carried out in order to obtain reliable and complete data. These farms are described in García-Gudiño et al. [8].

#### 2.2.2. Step 2: LCA Calculation and Climate Change for DMU's

The second step is the estimation of the environmental impact of Iberian pig farms through the LCA methodology. Environmental impacts were estimated using models, emission factors, and databases previously developed, without direct measurement of emissions [36–38]. The systems evaluated were farrow-to-finish systems that fatten all the piglets produced on the farm, in order to evaluate potential impacts of the system rather than impacts directly related to the orientation of the system. Therefore, the functional unit was one kilogram of live weight at farm gate. Analyses of LCA were performed by Simapro software (version 8.5.2.0, PRé Consultants, Amersfoort, The Netherlands) and the ecoinvent v3.1 database [39] for background data related to transportation and electricity production.

The environmental impacts of Iberian pig production considered were climate change ILCD (CC, kg $CO^2$ eq.) and land occupation (LO, $m^2$.year). Environmental variables employed were derived from García-Gudiño et al. [8].

#### 2.2.3. Step 3: Eco-Efficiency Assessment Using DEA Model

The third step uses the DEA model to assess efficiency and determine best practices or benchmarks as well as targets for inefficient DMUs. For this purpose, we follow the literature on the analysis of production efficiency through DEA [9,40,41]. This method considers the substitution possibilities between inputs and outputs by estimating efficient production frontiers from the data of a number of production units. In our study, the evaluation of eco-efficiency is estimated using a multi-objective DEA model oriented to

the value of pig production (EUR/year) that minimizes the undesirable output, that is, the environmental impact on climate change (CC, kg $CO_2$ eq.). The value of pig production (EUR/year) is considered the income from the sale of pigs in a year. The model based on variable returns to scale (VRS) assumption considers producers' differences in size and scale, as discussed by Lozano et al. [21]. Furthermore, Coelli et al. [40] indicate that most livestock production systems operate under the VRS assumption.

The measurement of technical efficiency is mainly based on two approaches: on the one hand, the parametric methodology that includes the construction of a stochastic [42] or deterministic frontier [43–45], and, on the other hand, the non-parametric methodology. Currently, the assessment of technical efficiency using the non-parametric approach was carried out using the data envelopment analysis (DEA), a method that uses linear programming to calculate an envelope or frontier from the available data of a set of production units or DMU so that the envelope is determined by the efficient units, while those that are not found in the envelope are considered inefficient [46–48].

The use of DEA methodology allows for identifying best practices or benchmarks for each inefficient DMU, new levels of production and CC, and necessary changes in input or resource levels. As these benchmarks define the targets, the inefficient DMU must follow its best operational/management practices to achieve the target in order to reach an eco-efficient state [35]. All targets are positioned on the efficient frontier; for instance, if inefficient DMUs achieve one of their alternative targets, they will become efficient. The general multi-objective DEA model, which considers VRS assumption and its constraints, can be presented as the following:

$$\text{Max } \varnothing_1 \ldots \varnothing_s \tag{1}$$

$$\text{Min } \varphi_1 \ldots \varphi_m \tag{2}$$

Subject to

$$\sum_{j=1}^{n} y_{rj}\, \lambda_j \ \geq\ \varnothing_r\, y_{r0};\ \forall\, r = 1,\, \ldots,\, s \tag{3}$$

$$\sum_{j=1}^{n} x_{ij}\, \lambda_j \ \leq\ \varphi_i\, x_{i0}\,;\ \forall\, i = 1,\, \ldots,\, m \tag{4}$$

$$\sum_{j=1}^{n} \lambda_j = 1 \tag{5}$$

$$\varnothing_r \geq 1;\ \forall\, r = 1,\, \ldots,\, s \tag{6}$$

$$\varphi_i \leq 1;\ \forall\, i = 1,\, \ldots,\, m \tag{7}$$

$$\varnothing_r\, \varphi_i, \lambda_j \geq 0 \tag{8}$$

where $m$ is the input, $s$ is the output and $n$ DMUs, $x_{ij}$ is the input $i$ of DMU $j$, $i = 1, \ldots, m$; $y_{rj}$ is the output $r$ of DMU $j$, $r = 1, \ldots, s$, with $j = 1, \ldots, n$; $\lambda_j$ is the contribution intensity of best practice or benchmark $j$ to the DMU target under evaluation. Increasing outputs ($\varnothing_r$) and decreasing inputs ($\varphi_i$) are optimised, while constraints ensure that these new levels are on the eco-efficient frontier. The constraint $\sum_{j=1}^{n} \lambda_j = 1$ guarantees VRS of the model.

The main advantage of combining DEA and LCA in the same methodological framework is that it allows the simultaneous optimisation of a production system's environmental impact and performance through competitive benchmarking processes. In other words, this approach provides reference pairs according to eco-efficiency criteria for the farms that are part of the production system [34].

### 2.2.4. Step 4: Determinants of Eco-Efficiency: Tobit Model

DEA has limited use for identifying the drivers of inefficiency [18]. This problem has usually been solved by further analysis with a deeper exploration of the factors hy-

pothesised to be related to inefficiency using Tobit models or truncated regression techniques [11,24,46]. The aim was to discover why some production units are more efficient than others. Numerous studies show that the most common explanation is differences in social and structural aspects, as well as in management capacity and decision-making processes [11,18,24,46]. For Ceyhan et al. [49], the appropriate regression for the level of eco-efficiency as the dependent variable is a Tobit regression.

The Tobit model also called the censored regression model, is used to estimate linear relationships between variables when there is left or right censoring in the dependent variable. The model is defined as follows:

$$\text{EE} \geq \beta X_k + \varepsilon_k \tag{9}$$

where EE is each calculated eco-efficiency score, $\varepsilon_k \sim N(0, \sigma^2)$ and $\beta$ is the vector of model parameters for the vector of explanatory variables $X_k$ [50].

In this study, the Tobit model is used to analyse the effect of socio-demographic aspects of the producer (family size, civil state, age, experience, education level, annual work unit) and structural and management characteristics of the farm (percentage owned area, percentage small ruminant livestock units, percentage of the area used for livestock, protected designations of origin (PDO), type of management and the level of *montanera* orientation on the eco-efficiency levels of the pig production system. These possible determinants of eco-efficiency were collected through face-to-face questionnaires mentioned in Section 2.1 and are detailed in Table 1. The variable level of *montanera* orientation has been categorised according to the proportion of pigs fattened in *montanera*: high, medium, and low. The medium level of fattening dedication was set in the range $(\overline{x} - \frac{1}{2}SD, \ \overline{x} + \frac{1}{2}SD)$, where $\overline{x}$ is the mean value, and SD is the standard deviation [45,51].

**Table 1.** Definition of dependent and explanatory variables of the inefficiency used in Tobit models.

| Dependent Variables | Definition of the Variables |
|---|---|
| | Social and demographic aspects |
| Family size | Number of family members |
| Number of children | Number of children |
| Civil state | Dummy = 1 If the producer is married, 0 if he is single |
| Age | Manager age |
| Experience | Number of years of managerial experience |
| Education level | Dummy = 1 If it is secondary or higher, 0 if it is primary level or without studies |
| AWU | Annual work unit |
| | Farm and management characteristics |
| % owned area | Own area as a percentage of total area |
| % small ruminant livestock units | Percentage of small ruminant livestock units |
| % of the area used for livestock | Percentage of land area used by livestock |
| PDO [a] | Dummy =1 if the products of animal origin belong to PDO; 0 if they do not. |
| Type of management | Dummy =1 if it is an extensive management, 0 if it is not (intensive management) |
| Level of *montanera* orientation | Proportion of pigs fattened in *montanera* |

*Note: "Eco-efficiency level" spans all Dependent Variables rows.*

[a] PDO: Protected Designations of Origin (PDO).

### 2.3. Statistical Analysis

All statistical analyses were carried out with SPSS for Windows software (v.16.0, SPSS Inc., Chicago, IL, USA) [52], while the program used to calculate the DEA model was the deaR package version 1.2.1 for R [53]. Finally, Eviews version 11 [54] was used to determine the censored regression model, the Tobit model.

## 3. Results

### 3.1. Description of the Iberian Pig Production System

Table 2 shows the main characteristics in relation to Iberian pig production and the *dehesa* ecosystem in farms participating in this research. The data collected show that the main activity of the farms is the fattening management, based on the use of acorns and other natural resources of the *dehesa*, although there was a high variability in the data in terms of surface area, the number of breeders and number of pigs produced (Table 2).

**Table 2.** General information of the participant Iberian pig farms (*n* = 35).

|  | **Mean** | **Standard Deviation** | **Min.** | **Max.** |
|---|---|---|---|---|
| Total surface (ha) | 646.4 | 627.00 | 28.50 | 3000 |
| Surface of *dehesa* (ha) | 498.00 | 437.80 | 18.00 | 2000 |
| % Surface used of *dehesa* | 84.00 | 25.89 | 0 | 100 |
| Number of sows per farm | 27.60 | 25.75 | 0 | 100 |
| Number of reproductive males per farm | 0.89 | 1.08 | 0 | 4.20 |
| Number of piglets fattened per farm | 319.70 | 315.00 | 0 | 1260 |
| CC [a] (kg $CO_2$ eq./kg LW) | 3.70 | 0.69 | 2.87 | 6.07 |
| LO [b] ($m^2$.year/kg LW) | 39.42 | 21.49 | 13.83 | 126.0 |

[a] CC: Climate change. [b] LO: Land Occupation.

### 3.2. Eco-Efficiency Assessment Using LCA–DEA Approach and Tobit Model

#### 3.2.1. Variables

The multi-objective DEA model is built from two inputs and two outputs (Table 3). As inputs, the use of surface area and the number of reproductive females were chosen because they are the main control factors affecting eco-efficiency, according to the characteristics of the Iberian pig production developed in the *dehesa* [26]. As outputs, the porcine production value was used, including the main product and by-products of each pig sold. The environmental impact on CC was calculated by García-Gudiño et al. [8] through the LCA methodology, considering previous literature [35,41].

**Table 3.** Variables used in the data envelopment analysis (DEA) model (*n* = 35).

|  | **Outputs** | | **Inputs** | |
|---|---|---|---|---|
|  | **Production Value (EUR)** | **Climate Change (CC, kg $CO_2$-eq/kg LW)** | **Surface in *Montanera* (ha)** | **Number of Sows** |
| Mean | 129,338 | 3.7 | 498.65 | 27.6 |
| SD [a] | 112,458 | 0.69 | 442.65 | 25.75 |
| Minimum | 13,502 | 2.87 | 0 | 0 |
| Maximum | 634,500 | 6.07 | 2000 | 100 |

[a] Standard deviation.

#### 3.2.2. Eco-Efficiency Results

The multi-objective DEA model for an Iberian pig farm (DMU) is presented as following:

$$\text{Max } \varnothing_V \tag{10}$$

$$\text{Min } \varnothing_{CC} \tag{11}$$

Subject to

$$\sum_{j=1}^{n} y_{Vj}\, \lambda_j \ \geq\ \varnothing_V\, y_{V0} \tag{12}$$

$$\sum_{j=1}^{n} y_{CCj}\, \lambda_j \ \leq\ \varnothing_{CC}\, y_{CC0} \tag{13}$$

$$\sum_{j=1}^{n} x_{Mj} \, \lambda_j \; \leq \; x_{M0} \tag{14}$$

$$\sum_{j=1}^{n} x_{Sj} \, \lambda_j \; \leq \; x_{S0} \tag{15}$$

$$\sum_{j=1}^{n} \lambda_j = 1 \tag{16}$$

$$\varnothing_V \geq 1 \tag{17}$$

$$\varnothing_{CC} \leq 1 \tag{18}$$

$$\varnothing_{V,} \; \varnothing_{CC}, \; \lambda_j \geq 0 \tag{19}$$

where the inputs correspond to surface area in *montanera* (*M*) and the number of sows (*S*), and the outputs correspond to the economic value of pig production (*V*) and environmental impact on CC. A VRS model is assumed to consider differences in size and scale.

There are several approaches to dealing with undesirable output (CC), in Rebolledo-Leiva et al. [34], the function to maximise includes the inverse of the undesirable output because maximising the inverse is equivalent to minimising it, while in Lozano et al. [21] the undesirable output is treated as an input to minimise. This model has two objectives; on the one hand, the economic value of pig production is maximised, while the undesirable output of CC is minimised, all while keeping the level of inputs constant. The advantage of using independent objective functions, such as in our study, is that it allows finding goals considering these two objectives simultaneously: maximising the production value and minimising the environmental impact on CC [35].

The DEA matrix (Table S1 of the Supplementary Material) was applied in the optimisation model to calculate the average of eco-efficiency values and benchmark values. Table S1 also presents the eco-efficiency value calculated for all pig farms. On average, the Iberian pig farms in the *dehesa* showed a high level of eco-efficiency. The average estimated level was 0.919, suggesting that the average farm could decrease its environmental impact by 8% by CC, given the level of inputs and production technology when farms adopt the observed best practices. The minimum and maximum eco-efficiency score was estimated at 0.479 and 1, respectively. These observed differences between the minimum and maximum values indicate a considerable degree of variation in the eco-efficiency of the *dehesa* Iberian pig systems.

The frequency distribution of the eco-efficiency estimates obtained is presented in Table 4. There is evidence that there is some variation in the use of existing technology in terms of eco-efficiency. Fourteen of the thirty-five farms, i.e., 40% of the total, are fully efficient from a technical and environmental point of view, revealing that the Iberian pig farms in the *dehesa* are using the current technology fairly rationally in terms of management. The highest number of inefficient farms was found in the ranges 0.80 to 0.90 and 0.90 to 0.99, with nine farms each, and the lowest in the score range from 0 to 0.80 with three farms. A total of 14 farms had eco-efficiency values of 1 constituting 40% of the sample.

**Table 4.** Frequency distribution of farms, by eco-efficiency estimates from the data envelopment analysis (DEA) model (*n* = 35).

| Level of Eco-Efficiency | Number of Farms | % | Mean |
|---|---|---|---|
| Low < 0.80 | 3 | 8.6 | 0.64 |
| Medium 0.8–0.9 | 9 | 25.7 | 0.86 |
| High 0.9–0.99 | 9 | 25.7 | 0.94 |
| Eco-efficient [a] | 14 | 40 | 1 |
| Total | 35 | 100 | 0.919 |

[a] Eco-efficient indicates a level of eco-efficiency of 1.

Figure 3 represents the distribution of mean total pig production (kg/year) and LO values according to the level of eco-efficiency achieved.

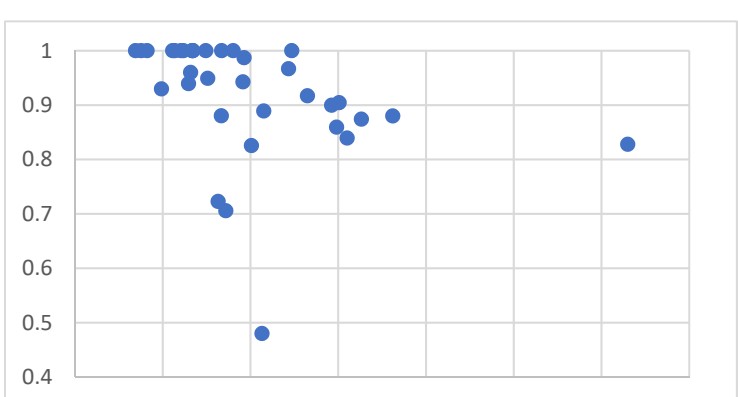

**Figure 3.** Eco-efficiency level (*y*-axis) with respect to land occupation (LO) (*x*-axis).

3.2.3. Targets for CC, Output Value and Inputs for Inefficient DMUs

Once the eco-efficiency value were determined, targets for the variables for inefficient pig farms were calculated. The DEA model provides targets for inputs (surface area in *montanera* and number of sows), which allow reducing current CC levels while maximising the value of pig production to become eco-efficient.

Table S2 in the Supplementary Material presents the slack and CC reduction targets, as well as the target for increasing the value of pig production and its percentage, concerning the original values considered in the analysis of inefficient farms.

For inefficient farms, important reduction targets are proposed. Inefficient farms, if they adopt the best practices of their benchmarks, can reduce the surface area in *montanera* used by 11.98 % and the number of sows by 13.54 %. On the other hand, the economic value of pig production (EUR/year) can be increased by 18.7% while reducing the environmental impact of the CC of the system by 8.3% (Table 5).

**Table 5.** Average percentage reduction of surface used in *montanera*, sows, climate change, and increase in production value.

|  | Value |
|---|---|
| Reduction percentage of surface in *montanera* | 11.98 |
| Reduction percentage of number of sows | 13.54 |
| Climatic Impact on CC reduction | 8.32 |
| Production value increase | 18.68 |

The method developed in this study allows us to know the intensity (λ) of each benchmark or best practice of each inefficient pig farm, i.e., what is called benchmarking. These intensities for each inefficient DMU are presented in Table S3. The information in this table provides guidelines for determining an improvement plan for inefficient DMUs to become efficient. The best practices in our sample are DMU 30, DMU 27 and DMU 23, which are used as benchmarks for a total of 41 inefficient farms. These farms are characterised by traditional management. These farms are focused on producing Iberian pigs fed on a natural resources-based diet. The number of breeders is adapted to the maximum number of Iberian pigs that can be fattened on the farm according to the size of the surface and density of trees. Reproductive management is traditional, natural mating and two farrowing per year. Breeding is carried out extensively until the *montanera* season, when the animals are fattened only with natural resources.

To illustrate this procedure, we take DMU 3 as an example. This farm has an eco-efficiency level of 0.84 and a CC target of 2.94 kg $CO_2$ eq. and its reference points are DMU

27 and DMU 30 with intensities of 0.1745 ($\lambda_{27}$) and 0.8255 ($\lambda_{30}$), respectively. Tables S1–S3 show the DEA matrix, inputs/outputs targets, and the benchmarks intensities, respectively, for DMU 3 and other pig inefficient farms.

3.2.4. Determinants of Inefficiency

The effect of the factors hypothesised to influence inefficiency assessed using the Tobit model are shown in Table 1. The results of the regression analysis are shown in Table 6.

**Table 6.** Statistical analysis of Tobit model.

| Variable | Coefficient | Std. Error | z-Statistic | Prob. |
|---|---|---|---|---|
| Social and demographic aspects | | | | |
| Family size | −0.002883 | 0.035695 | −0.080764 | 0.9356 |
| Number of children | 0.065596 | 0.020803 | 3.153.233 | 0.0016 |
| Civil state | −0.026756 | 0.046468 | −0.575791 | 0.5648 |
| Age | 0.007590 | 0.005261 | 1.442.852 | 0.1491 |
| Experience | −0.011463 | 0.004038 | −2.839.121 | 0.0045 |
| Education level | −0.138727 | 0.050600 | −2.741.620 | 0.0061 |
| AWU | 0.032217 | 0.052929 | 0.608680 | 0.5427 |
| Farm and management characteristics | | | | |
| % owned area | 0.001732 | 0.000867 | 1.996.838 | 0.0458 |
| % small ruminant livestock units | 0.054434 | 0.124319 | 0.437859 | 0.6615 |
| % of the area used for livestock | 0.005509 | 0.001689 | 3.261.379 | 0.0011 |
| PDO | 0.050314 | 0.045072 | 1.116.291 | 0.2643 |
| Level of *montanera* orientation | 0.081780 | 0.022082 | 3.703.435 | 0.0002 |
| Type of management | 0.084327 | 0.048146 | 1.751.482 | 0.0799 |
| Constant | 0.079468 | 0.011470 | 6.928.203 | 0.0000 |
| Log likelihood | | 26.723 | | |
| AIC | | −1.0602 | | |

It was assuming VRS model how the best fit (Log-likelihood = 26.723; AIC = −1.0602), the crucial determinants that positively affected eco-efficiency in Iberian pig farms were the number of children, the percentage of own surface area, the percentage of livestock use, and the high proportion of pigs fattened in *montanera*. On the other hand, years of farm activity and educational level negatively affected eco-efficiency levels ($p < 0.05$).

## 4. Discussion

Iberian pig production is a heterogeneous system due, on the one hand, to the fact that the volume of pig production is linked to the area of available *dehesa*. On the other hand, the variability in the number of sows reflects the different intensity with which breeding is practised, from full-cycle farms that exclusively fatten the piglets produced, although with different intensities of use of the pasture, to farms where the supply of piglets for other farms is a management goal [27]. The data obtained in terms of surface areas and pig censuses are close to other studies carried out in the *dehesa* ecosystem [26,31,55].

Regarding CC, the data obtained indicate that Iberian pig production is close to traditional pig production [56], but it has greater LO impacts than other pig breed systems [25,56]. It could be mainly explained by the large surface area required for fattening animals fed exclusively on natural resources from the *dehesa* [57].

The approach's selection of inputs and outputs has been performed to reflect the pig production process developed in the *dehesa* synthetically. In addition, previous studies were taken into account to analyse the eco-efficiency of agricultural and livestock enterprises [4, 11,18,22,24,35,46]. Furthermore, the rule shown by Cooper et al. [58] has been considered to not excessively limit the model's degrees of freedom. The recommendation is to select a value of *n* that satisfies n ≥ {m × s; 3(m + s)} where *n* is the number of DMUs (35 pig farms in this study), *m* is the number of inputs, and *s* is the number of outputs. Therefore, the

number of DMUs in our sample satisfies the rule in this study and the requirements for this methodology were met.

The level of eco-efficiency obtained is in line with previous studies on the environmental impact assessment of pig production systems, where it is highlighted that lower environmental impacts can be achieved in pig production linked to the territory using native breeds [26]. This is due to a lower dependence on off-farm feed due to the feeding strategy of these production systems with greater use of natural resources, such as acorns available in the meadows and pastures of the *dehesa* [8,25,59,60].

Regarding Figure 3, we can conclude that the increase in LO implies a decrease in eco-efficiency levels. This is probably due to the increase in the number of pigs fattened with feedstuffs, thus increasing the hectares needed to produce raw materials for feed production. Another reason could be the low density of holm oaks and cork oaks, which impacts the surface area needed in the *montanera* [57]. Reforestation techniques in the *dehesa* could positively impact eco-efficiency levels [61].

The objective of reducing the area of *montanera* while achieving the same production is possible with reforestation [61]. An increased wooded area can be used to fatten more pigs with natural resources (acorns and grass), reducing the environmental impact and, at the same time, increasing the income as the pigs fattened with acorns have a higher commercial value. In addition, the reduction in the number of reproductive females in inefficient farms is a fact other authors have shown in the *dehesa* [29]. This is more evident for the management of multi-output systems (pig fatteners *montanera*, pig fatteners *cebo campo*, piglet sales).

These analysis projections reveal the maximum potential for input and environmental impact reduction that can be achieved in Iberian pig production in the *dehesa*. There are no existing studies on Iberian pigs, but our sample has a better projection of improvement than other previously evaluated livestock systems, with reductions in environmental impact of more than 30% [4,62,63]. Furthermore, what the above projections confirm is that it can be concluded that actions are needed to improve economic rather than environmental performance since traditional Iberian pig production systems are associated with sustainable productions based on natural resources and low environmental impact [8,26,59]. Possibly these actions should be aimed at decreasing dependency on external inputs such as feedstuffs. Production systems based on fattening *montanera* produce better environmental and economic benefits [8,33]. On the contrary, those based on fattening *cebo campo* produce pigs fed with significant quantities of compound feed and a product of poorer commercial quality according to the Spanish regulations regarding the quality of Iberian pork products [57]. However, assessing the economic sustainability of pig farms is a complex problem, as many short- and long-term factors are involved [64].

The crucial determinants related to social and demographic aspects that positively affected eco-efficiency in Iberian pig farms was the number of children. In contrast, the variable number of years of activity and educational level, contrary to expectations, negatively affected eco-efficiency. The latter may be because more educated owners pursue higher profitability production on their farms and thus move away from a traditional production model, which, as we have found in our study, leads them to be more eco-efficient. This could also be because experienced farmers are more reluctant to change their management habits. Li et al. [46] indicated in a study with 773 pig farms that the years of experience and dedication to the activity had a negative effect. Also, other studies focused on agriculture indicated that higher education and specialised training affected efficiency improvement [17]. Although there is some controversy with the educational level, numerous studies indicated a positive relationship with university education, mainly because more education may imply more adaptation to new market opportunities, distancing from a traditional production model [65]. While other studies have found an influence of age on eco-efficiency, our model did not detect any influence.

In terms of management, farm characteristics such as land ownership, livestock use, and the high proportion of pigs fattened in *montanera* positively affected the level of

eco-efficiency. The three factors mentioned are closely related to Iberian traditional pig production in the *dehesa* where the use of natural resources is essential for developing production. There are two types of fattening on the *dehesa*, *montanera* and *cebo de campo*. Our study shows that those farms that perform traditional management are more eco-efficient, according to results revealed by Horrillo et al. [26] and García-Gudiño et al. [27], mainly by optimal use of natural resources provided by the *dehesa* ecosystem.

PDO certification is a quality indicator and an instrument that reduces the asymmetry of information between producer and consumer, specifying the production system [33]. Contrary to expectations, PDO did not affect eco-efficiency, as indicated by a study by García-Cornejo et al. [24] on livestock farms in northern Spain. In our study, it could be explained by the fact that the production of Iberian pigs is already nationally regulated [57].

Although this method used a robust methodology to calculate eco-efficiency scores and a Tobit regression approach, the small sample size probably limited our ability to identify the most statistically significant variables. The study of the Iberian pig production systems is complex because there are different types of fattening of the animals (*montanera* and *cebo campo*) and in order to obtain reliable and complete data it is necessary to visit the same farm several times. Farm visits could not be carried out in the same year for all the farms, so the study was carried out over three consecutive years (2016–2018). This fact adds a bias which, in principle, should not be relevant because prices of inputs and outputs remained stable, but should be taken into account for further studies. Future studies should emphasise larger samples of production units from different locations to understand better the role of other factors, such as management, information use, and decision-making process [66,67]. Despite these limitations and the sample size, our study has contributed to the existing literature as the first study on eco-efficiency in Iberian pig farms in the *dehesa* ecosystem.

## 5. Conclusions

In this study, an evaluation of the eco-efficiency of Iberian pig farms in the *dehesa* area of the Iberian Peninsula has been carried out. The application of a combined LCA and DEA methodology of Iberian pig farms based in the *dehesa* area of the Iberian Peninsula has proved to be a valuable tool for the comparative assessment of environmental, technical, and economic parameters. The Iberian pig farms in the *dehesa* showed a high level of eco-efficiency, suggesting that the average farm could decrease its climate impact given the level of inputs and production technology, provided that the farms adopt the observed best practices. The farmer's professionalism and profile influences eco-efficiency. Other farm characteristics related to natural resource use, and the proportion of pigs fattened in *montanera* affected the level of eco-efficiency.

The production of Iberian pigs following traditional management systems is more eco-efficient and has a lower environmental impact, increasing its impact as it moves towards a fattening system in *montanera*. The reduction of environmental impact implies a reduction in the consumption of feedstuffs. Therefore, better management of natural resources and adapting production to the type of farm could reduce feedstuff dependency and make traditional Iberian pig production more environmentally friendly and eco-efficient.

Finally, it should be noted that the results of our study are of interest to stakeholders and policymakers to identify the most environmentally friendly practices to optimize resources on Iberian pig farms. Policies should be aimed at promoting the system of production of Iberian pigs in the *montanera*. The results can be used to implement cleaner production strategies that reduce national emissions. Furthermore, future research could be carried out to evaluate this eco-efficiency model in other species that grow in the *dehesa* in order to improve the conservation of this ecosystem.

**Supplementary Materials:** The following are available online at https://www.mdpi.com/article/10.3390/agriculture13010083/s1, File S1: Questionnaire carried out on participating farms, Table S1: Data Envelopment Analysis (DEA) matrix for the complete set of farms and eco-efficiency score.

Table S2: Input/output target and operational reduction percentages for inefficient farms. Table S3: Benchmarks intensities of inefficient farms.

**Author Contributions:** Conceptualization, E.A. and J.P.; methodology, E.A. and J.P.; formal analysis, E.A. and J.P.; investigation, E.A., J.P. and J.G.-G.; resources, I.B.-P., F.G.-L.; writing—original draft preparation, E.A.; writing—review and editing, E.A., J.P., J.G.-G., I.B.-P. and F.G.-L.; funding acquisition, I.B.-P. All authors have read and agreed to the published version of the manuscript.

**Funding:** This research was funded by National Institute for Agricultural and Food Research and Technology, grant number RTA2013-00063-C03-02.

**Institutional Review Board Statement:** Ethical review and approval were waived for this study, as our study was developed through farmer surveys.

**Data Availability Statement:** The data presented in this study are available on request from the corresponding author. The data are not publicly available due to project IP rules.

**Acknowledgments:** The authors would like to thank POD Dehesa de Extremadura, AECERIBER, ACPA and farmers for their support.

**Conflicts of Interest:** The authors declare no conflict of interest.

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
