# Peer review of "Targeting Environmental and Technical Parameters through Eco-Efficiency Criteria for Iberian Pig Farms in the dehesa Ecosystemâ€"

_agriculture, doi:10.3390/agriculture13010083_

Round 1
Reviewer 1 Report
A brief summary
This study is great as it provides a clear picture about the eco-efficiency of extensively reared pigs on natural resources and its impact on environment. In addition to benchmarks for improving inefficient farms but this part was not mentioned in the discussion section. This study could reflect the impact of livestock farming on climate change.
Specific comments
Abstract:
- Line 13-14: remove (analysis in), remove (of in depth debate)
- Line 14-15: write (pig pasture production systems) instead of (pig production systems reared on pasture)
- Line 22: farms not farmers
- Line 25: output or input
Keyword: add (farms) after (pig)
Introduction:
- Line 34: remove (in the primary sector) or mention its name
- Line 40: add (these) before (goals)
- Line 44: add (also) after (and)
- Line 59-60: like not with, that not in order to
- Line 61: add (also) before (developed)
- Line 64: remove (resulting)
- Line 70-75: not clear what you meant by this paragraph, please clarify
- Line 78: accepting that not acceptable
Methods:
- Ethical approval not present, Please add it
- File S1 not present
- Line 119: remove (as), remove (elsewhere), write (in) instead of (by)
- Line 120: collected not achieved, in not for , include not were
- Line 124: add (environmental) before (impacts)
- Line 130: add (of pig production) after (impact), write DMU meaning
- Line 138: remove
- Line 139: remove (subsequently)
- Line 193: was not will be
- Line 206: write UTH meaning
- Line 220: add a separate heading for the statistical analysis part
Results:
- Line 228-231:discuss Table 2
- Line 292: figure 3 is not mentioned in text, discuss
- Line 297: add (value) after eco-efficiency
- Line 315-316: â…„ not DMU, 18 is not seen in Table S3
- Line 323-327: not clear what you meant, please clarify
- Line 329-331: discuss Table 6
Discussion:
- Line 358: correct od to of
- Line 359: of not in
- Line 360: for not in, add (that is) before (linked)
- Line 362: strategy not strategic
- Line 365: labels is not correct for axis in figure 3
- Line 399: was not were
- Line 408: start a new statement with although
- Line 401-403: not clear meaning, please clarify
Author Response
The authors would like to thank the reviewers for their comments that have helped to improve the manuscript (agriculture-2050035). Following a detailed answer to each comment. We hope that this modified version of the manuscript will meet the expectations of the two reviewers and the editorial team.
All these comments are detailed in the attached file.

Author Response

(The authors gave the same response as above.)

Round 2
Reviewer 2 Report
The authors have addressed the comments.
Still a few things to be fixed:
- I understand you are not using a panel data, but this should be made clear in the abstract as well as in the text indicating why you consider three years.
-the notation of equations in section 3.2.2 should be consistent with the notation in equations in section 2.2. I think there are some disaligments.
- deep revision of English required. There are typos also in the added sentences.
Author Response
The authors would like to thank the reviewer 2 for their comments that have helped to improve the manuscript (agriculture-2050035). Following a detailed answer to each comment. We hope that this modified version of the manuscript will meet the expectations of the reviewer and the editorial team.
